# TRPM4 Blocking Antibody Protects Cerebral Vasculature in Delayed Stroke Reperfusion

**DOI:** 10.3390/biomedicines11051480

**Published:** 2023-05-19

**Authors:** Bo Chen, Shunhui Wei, See Wee Low, Charlene Priscilla Poore, Andy Thiam-Huat Lee, Bernd Nilius, Ping Liao

**Affiliations:** 1Calcium Signalling Laboratory, Department of Research, National Neuroscience Institute, Singapore 308433, Singapore; bo_chen@nni.com.sg (B.C.);; 2Health and Social Sciences, Singapore Institute of Technology, Singapore 138683, Singapore; 3Department of Cellular and Molecular Medicine, KU Leuven, 3000 Leuven, Belgium; 4Neuroscience Academic Clinical Programme, Duke-NUS Medical School, Singapore 169857, Singapore

**Keywords:** transient receptor potential channels, antibody, reperfusion injury, middle cerebral artery occlusion, model, vascular, protection, hypoxia, stroke, therapy

## Abstract

Reperfusion therapy for acute ischemic stroke aims to restore the blood flow of occluded blood vessels. However, successful recanalization is often associated with disruption of the blood-brain barrier, leading to reperfusion injury. Delayed recanalization increases the risk of severe reperfusion injury, including severe cerebral edema and hemorrhagic transformation. The TRPM4-blocking antibody M4P has been shown to alleviate reperfusion injury and improve functional outcomes in animal models of early stroke reperfusion. In this study, we examined the role of M4P in a clinically relevant rat model of delayed stroke reperfusion in which the left middle cerebral artery was occluded for 7 h. To mimic the clinical scenario, M4P or control IgG was administered 1 h before recanalization. Immunostaining showed that M4P treatment improved vascular morphology after stroke. Evans blue extravasation demonstrated attenuated vascular leakage following M4P treatment. With better vascular integrity, cerebral perfusion was improved, leading to a reduction of infarct volume and animal mortality rate. Functional outcome was evaluated by the Rotarod test. As more animals with severe injuries died during the test in the control IgG group, we observed no difference in functional outcomes in the surviving animals. In conclusion, we identified the potential of TRPM4 blocking antibody M4P to ameliorate vascular injury during delayed stroke reperfusion. If combined with reperfusion therapy, M4P has the potential to improve current stroke management.

## 1. Introduction

An ischemic stroke occurs when the blood supply to part of the brain is interrupted or reduced. Insufficient blood supply to the brain results in a limited supply of oxygen and other nutrients to meet tissue metabolic demands, leading to brain damage [1]. Current management for acute ischemic stroke is mainly focused on reperfusion therapy by restoring blood flow to the affected brain tissues. Reperfusion therapies for acute ischemic stroke include pharmacological thrombolysis via the application of tissue plasminogen activator (tPA) and mechanical thrombectomy [2]. The oxygen and nutrients following successful reperfusion thus salvage the brain tissues within the penumbra region that are dysfunctional but not yet dead. Paradoxically, restoring blood supply may injure the brain tissue, which is known as reperfusion injury [3]. Clinical outcomes following reperfusion therapy are often confounded by reperfusion injury. The key pathophysiological change during reperfusion injury is the damage to the blood-brain barrier (BBB) [4]. Disruption of the BBB sometimes results in severe consequences such as vasogenic edema and hemorrhagic transformation. Importantly, the severity of reperfusion injury increases over time, which determines the time window for therapy. The gold-standard treatment of tPA is recommended to be administered to eligible patients given up to 4.5 h after symptom onset [5]. Mechanical thrombectomy is best for patients suffering from a major stroke with a large vessel occlusion. The guidelines recommend thrombectomy to be given 6 to 16 h from the last seen well. Beyond these time windows, the risks of side effects from recanalization outweigh the benefits, causing more morbidity and mortality [6]. Due to the limitation of these time windows, many stroke patients are unable to receive the potent reperfusion treatment because they do no not reach the hospital soon enough [7]. It is thus critical to extend the therapeutic time window to improve reperfusion therapy. How to protect BBB beyond the current time window has become a major challenge for stroke research.

Recently, the transient receptor potential (TRP) channels have emerged to play an important role in stroke pathophysiology [8]. The TRP channels constitute a superfamily that includes at least nine subfamilies: TRPP (polycystin or polycystic kidney disease), TRPML (mucolipin), TRPA (ankyrin), TRPV (vanilloid), TRPVL (vanilloid-like), TRPC (canonical), TRPN (nompC, or no mechanoreceptor potential C), TRPM (melastatin) and TRPS (soromelastatin) [9]. Among the TRP superfamily, the transient receptor potential melastatin-like subfamily member 4 (TRPM4) has recently emerged as an important drug target for stroke therapy [10,11,12,13,14,15,16,17] and many other diseases [18,19,20]. TRPM4 is a nonselective cation channel, conducting monovalent ions such as sodium [21]. Importantly, TRPM4 is activated by ATP depletion and an increase of intracellular Ca^2+^, which are important pathological features associated with hypoxia. Under hypoxic conditions such as stroke, TRPM4 activity is greatly enhanced [11,15,16,22]. Furthermore, TRPM4 expression is upregulated in surviving neurons and vascular endothelial cells close to the infarct core [11,22]. As a result of TRPM4 activation, sodium influx induces cell swelling and leads to oncotic cell death in neurons and vascular endothelial cells. Accordingly, blocking TRPM4 attenuates oncotic cell death [15]. The effect of TRPM4 on reperfusion injury is prominent. In an animal model of stroke reperfusion, MRI and PET scans show that TRPM4 inhibition by siRNA resulted in a drastic reduction of cerebral edema and infarction [11]. In a chronic hypoxia model, the application of TRPM4 siRNA was shown to improve spatial memory impairment and hippocampal long-term potentiation deficit [23].

As siRNA acts at the transcriptional level, it must be administered prior to protein upregulation. In an animal stroke model, we found that TRPM4 expression is upregulated as early as 2 h after middle cerebral artery occlusion [11]. Therefore, the application of TRPM4 siRNA beyond 2 h following stroke onset is unlikely to achieve optimal outcomes. It is best to use an antagonist to block the channel directly. However, current available TRPM4 blockers have various limitations, such as lack of specificity, requiring associated subunits, or toxicity [10]. TRPM4 has been reported to interact with sulfonylurea receptor-1 (Sur1) to form a SUR1-TRPM4 channel complex. SUR1 blockers sulfonylureas were shown to inhibit SUR1-TRPM4 function [24]. SUR1 is an auxiliary subunit of the K_ATP_ channel, which senses ATP levels in pancreatic β cells [25]. Sulfonylureas such as glibenclamide are widely used to control blood glucose levels in diabetic patients by regulating insulin secretion. As sulfonylureas are available in clinical practice, multiple trials of glibenclamide were carried out in stroke patients with or without diabetes mellitus. In some studies, the application of sulfonylureas before or after stroke onset reduced hemorrhagic transformation and attenuated cerebral edema with improved neurological outcomes [26,27,28,29,30]. Other retrospective studies on diabetic patients who later developed stroke revealed that sulfonylureas treatment achieved a similar outcome as other antidiabetic therapies [31,32,33,34,35]. Such controversies may arise from differences in patient inclusion criteria, dose of sulfonylureas, or the severity of diabetes mellitus. For example, the dose of glibenclamide was low in the study on stroke as higher doses could induce hypoglycemia in patients [36]. Another study showed that application of sulfonylurea glimepiride achieved neuroprotection against stroke only in normal mice but not in type 2 diabetic mice [37]. This result suggests that the presence of diabetes may be a confounding factor when sulfonylureas are used to manage stroke.

In view of the challenges among current TRPM4 blockers, we have developed a TRPM4-specific blocking antibody M4P [10]. M4P could bind to the TRPM4 channel from the extracellular space and inhibit channel function. In an early 3-h stroke reperfusion animal model, the application of M4P successfully ameliorates reperfusion injury [10]. In this study, we aim to examine the therapeutic effect of M4P during delayed stroke reperfusion. We hypothesize that TRPM4 blocking antibody could reduce delayed stroke reperfusion injury and possibly extend the time window of current reperfusion therapy.

## 2. Materials and Methods

### 2.1. Rat Middle Cerebral Artery Occlusion (MCAO) Model and Experimental Protocol

This study was approved by the Institutional Animal Care and Use Committee of Lee Kong Chian School of Medicine from Nanyang Technological University and National Neuroscience Institute. All experiments were performed following the ARRIVE guidelines (Animal Research: Reporting In Vivo Experiments) and in compliance with the NACLAR guidelines (National Advisory Committee for Laboratory Animal Research) of Singapore. A total of 300 adult male Sprague–Dawley rats, weighing 250–300 g, were divided into 4 groups: sham (*n* = 24), permanent MCAO (*n* = 20), 3 h transient MCAO (*n* = 57), and 7-h transient MCAO (*n* = 199). The animals were housed with a temperature maintained at around 23 °C, and a 12/12-h light/dark cycle was set. Pelleted food and water were available for the animals. The animals were monitored on a daily basis. Allocation of animal treatment was randomized by rolling a dice. All researchers involved in the study were blinded to the intervention. MCAO was induced as previously described [38]. Briefly, rats were anesthetized with ketamine (75 mg/kg) and xylazine (10 mg/kg) intraperitoneally. The rats were placed supine, a midline incision was made in the neck, and the left external carotid artery (ECA), internal carotid artery (ICA), and common carotid artery (CCA) were dissected. A silicon-coated filament (0.37 mm, Cat #403756PK10, Doccol Corp, Redlands, CA, USA) was inserted into the left ICA through ECA. A Laser-Doppler flowmetry (moorVMS-LDF2, Moor Instruments Inc., Wilmington, DE, USA) was used to measure cerebral blood flow. Animals with less than 70% cerebral blood flow reduction were excluded from the study. At 3 h or 7 h after occlusion, the suture and CCA vessel clip were removed, and the ECA was closed. One hour before recanalization, a single dose of 100 μg of antibody (M4P or control rabbit IgG) was injected intravenously via the tail vein. The production of GST-tagged antigens and the generation of M4P have been described previously [10,39]. For permanent MCAO, the filament was left inside ICA. For sham-operated animals, the same anesthetic procedure was applied, but no filament was inserted into the ICA. During the operation in all animals, heart rate, blood pressure, and rectal temperature were monitored using a data acquisition system PowerLab 4/35 from AD Instruments (AD Instruments, Dunedin, New Zealand). The body temperature was maintained at 37 °C ± 0.5 °C with a warm pad throughout the operation. The mortality rate for permanent MCAO was 10% (2/20); for sham-operated animals, it was 8.3% (2/24); for 3-h stroke reperfusion was 15.7% (9/57), and for 7-h stroke reperfusion (including animals used for immunostaining, TTC staining, and functional analysis) was 49.7% (99/199).

### 2.2. Cerebral Blood Flow Measurement

The midline scalp was incised to reach the cranial fascia. A subsequent left paramedian incision was made to the cranial fascia for the left MCAO. The skull bone was exposed by blunt dissection, and the bone area was prepared for applying a probe holder. After applying a 3% hydrogen peroxide solution to disinfect and dry the skull surface, the probe holder and the probe were placed on the skull surface at −1 mm from bregma, 5 mm lateral to the midline [40]. The Laser Doppler probe was purchased from Moor Instruments. The cerebral blood flow during the operation was monitored using the moorVMS-LDF laser Doppler monitor (Moor Instruments Inc., Wilmington, DE, USA).

### 2.3. 2,3,5-Triphenyltetrazolium Chloride (TTC) Staining and Evans Blue Extravasation

TTC staining was performed 24 h after the operation to quantify infarct volume. Brains were collected after the animals were euthanized, and the cerebellum and overlying membranes were removed. Using a brain-sectioning block, the brains were sectioned into 2-mm-thick coronal slices. The brain sections were stained with 0.1% TTC (Sigma, St. Louis, MO, USA) solution at 37 °C for 30 min and then preserved in 4% formalin solution. The sections were scanned, and the infarct was captured with an image analyzer system (Scion image, Microsoft windows, Scion Corporation, Frederick, MA, USA). Calculation of edema-corrected lesion was performed as described previously [41].

Blood-brain barrier permeability was assessed by measuring Evans blue extravasations. Evans blue (E2129; Sigma-Aldrich) was prepared at a concentration of 2%. Half an hour before filament withdrawal, Evans blue was injected into the jugular vein at a dose of 4 mL/kg body weight. Three hours after reperfusion, rats were perfused transcardially with phosphate-buffered saline (PBS). After taking a picture, the ipsilateral and contralateral hemispheres were dissected, weighted, and homogenized in 1:3 weight (mg): volume (µL) ratios of 50% trichloroacetic acid (TCA) (T9159; Sigma-Aldrich, St. Louis, MO, USA) in saline. Following centrifugation at 12,000× *g* for 20 min, the supernatant was collected and thoroughly mixed with 95% ethanol (1:3) by repeated pipetting for fluorescence spectroscopy (620 nm/680 nm) using a Tecan infinite plate reader. Results were quantified according to a standard curve and presented as a µg of Evans Blue per gram of brain tissue.

### 2.4. Immunofluorescent Staining

The rat brains collected at different time points after MCAO were harvested and sectioned at 10 µm in thickness. Following fixation with 4% paraformaldehyde, the brain slice was incubated in 100 μL blocking serum (10% fetal bovine serum in 0.2% PBST) for 1 h. Cerebral vasculature was stained with primary antibody anti-vWF (AB7356, 1:200, Milipore, Burlington, MA, USA), followed by Alexa Fluor 594 conjugated secondary antibody. Images were visualized by a confocal microscope (Fluoview BX61, Olympus, Tokyo, Japan). ImageJ was used to capture the change in vascular morphology. The total vascular area per image was quantified by ImageJ, and the average vascular diameter was determined using the shortest Feret diameter (Feret Min) as described previously [42].

### 2.5. Rotarod Test

Rotarod (Ugo Basile, Gemonio, Italy) was used to evaluate motor functions post-stroke. Before the operation, the rats received 3 training trials at 15-min intervals for 5 consecutive days. The rotarod was set to accelerate from 4 to 80 rpm within 10 min. The mean duration of time that the animals remained on the device 1 day before MCAO was recorded as internal baseline control. At different time points following surgery, the mean duration of latency was recorded and compared.

### 2.6. Electrophysiology

Whole-cell patch clamp was used to measure TRPM4 currents in HEK293 cells transfected with pIRES-EGFP-TRPM4 encoding mouse TRPM4 channel using Lipofectamine 2000. TRPM4 currents were recorded 24–48 h after transfection at room temperature (22–23 °C). Patch electrodes were pulled using a Flaming/Brown micropipette puller (Sutter Instrument) and polished with a microforge. Whole-cell currents were recorded using a patch clamp amplifier (Multiclamp 700B equipped with Digidata 1440A, Molecular Devices, San Jose, CA, USA). The bath solution contained (in mmol/L): NaCl 140, CaCl_2_ 2, KCl 2, MgCl_2_ 1, glucose 20, and HEPES 20 at pH 7.4. The internal solution contained (in mmol/lL): CsCl 156, MgCl_2_ 1, EGTA 10, and HEPES 10 at pH 7.2 adjusted with CsOH [20]. Additional Ca^2+^ was added to get 7.4 μM free Ca^2+^ in the pipette solution, calculated using the program WEBMAXC v2.10. Rabbit IgG or M4P was added into the bath solution at a concentration of 20 µg/mL half an hour before recording. ATP depletion was induced by applying a bath solution containing 5 mM NaN_3_ and 10 mM 2-deoxyglucose (2-DG) continuously through a MicroFil (34 Gauge, WPI Inc., Sarasota, FL, USA) around 10 µm away from the recording cells. The flow rate was 200 µL/min. The current–voltage relations were measured by applying voltage ramps for 250 ms from –100 to +100 mV from a holding potential of 0 mV. The sampling rate was 20 kHz, and the filter setting was 1 KHz. Data were analyzed using pClamp10, version 10.2 (Molecular Devices, San Jose, CA, USA).

### 2.7. Statistical Analysis

Data are expressed as the mean ± s.e.m. Statistical analyses were performed using GraphPad Prism version 6.0. Two-tailed unpaired student’s *t*-test was used to compare two means. One-way ANOVA with Bonferroni’s multiple comparison tests was used to compare ≥3 means. Two-way ANOVA with Bonferroni’s multiple comparison tests was used to analyze motor functions.

## 3. Results

### 3.1. Time-Dependent Vascular Injury Post MCAO

To understand the status of vascular health after stroke, MCAO was created in Sprague–Dawley rats. The brains were collected at different time points after occlusion: 3, 6, 9, 12, and 24 h. Immunostaining using an anti-vWF antibody on the ipsilateral hemispheres demonstrated time-dependent morphological changes in the cerebral vasculature (Figure 1A). Quantification of vascular staining revealed that the area of vascular staining decreases gradually after MCAO induction (Figure 1B). Loss of vascular staining indicates the degradation of vascular structure. At 6 h MCAO, the area of vasculature is lower than 3 h MCAO but still higher than those after 9 h MCAO. After 9 h MCAO, there was no change in the vascular areas, indicating that the vasculature loss had reached a maximal level. Vascular health is closely related to the reperfusion injury after stroke. Figure 1C compares two sample brains after 3 h MCAO reperfusion and 10 h MCAO reperfusion. The 10 h transient MCAO brain demonstrated a larger infarct area and a diffused hemorrhage at multiple sections within the infarct area.

### 3.2. M4P Inhibits TRPM4 Current

We have developed a TRPM4-blocking antibody, M4P, which inhibits rodent TRPM4 [10]. TRPM4 currents exhibit a typical outward rectifying property in HEK 293 cells transiently expressing mouse TRPM4 (Figure 2A). Compared to control rabbit IgG, the application of M4P significantly inhibited TRPM4 current (Figure 2B). TRPM4 is known to be activated by ATP depletion and increased intracellular calcium levels [43]. When ATP was depleted, a prominent higher TRPM4 current was observed in cells treated with control IgG (Figure 2C). Again, incubation with M4P significantly reduced TRPM4 current (Figure 2D). These results suggest that M4P suppresses TRPM4 current under both normoxic and hypoxic conditions.

### 3.3. M4P Reduces Mortality Rate and Infarct Volume in 7-h Stroke Reperfusion

We had reported that the application of M4P at 2 h post-MCAO could reduce reperfusion injury and improve functional recovery when recanalization was achieved at 3 h post-MCAO [10]. To examine the effect of M4P on delayed stroke reperfusion, we performed a 7-h transient MCAO model on rats (Figure 3A). M4P, control IgG, or vehicle was administered intravenously at 6 h post-MCAO. Reperfusion was achieved at 7-h MCAO by removing the filament. As a prolonged operation was performed, we evaluated first how this procedure affects animal mortality. In the permanent MCAO group, the animals received normal MCAO procedures without reperfusion; the mortality rate was 10% (2/20). The remaining animals all survived beyond 24 h. In a sham surgery group, animals received a similar 7-h operation procedure but without exposing carotid arteries. The mortality rate is 8.3% (2/24). In the 7-h transient MCAO group, the mortality rate is 15.6% (31/199) (Figure 3B). This result indicates that prolonged operational procedure with anesthesia increases mortality.

Next, we calculated the mortality rates in animals that recovered from 7-h operation of transient MCAO and observed for 14 days. In the vehicle treatment group and control IgG treatment group, the mortalities are similar at around 68–70%. In the M4P treatment group, the mortality rate decreases to 44.4% (8/18) (Figure 3C). Further analysis revealed that within the dead animals from the control IgG group, 76.5% (13/17) animals died within 24 h after occlusion, and the remaining 23.5% (4/17) animals died after 24 h post-operation. In M4P-treated animals, the proportion of animals that died after 24 h is 12.5% (1/8), lower than the control IgG group.

To evaluate the tissue damage, infarct volume was quantified in rat brains collected 24 h after occlusion (Figure 3D,E). The infarct volume of the M4P group was significantly lower than the vehicle and permanent MCAO groups. The control IgG group shows no difference from the vehicle and permanent MCAO groups. There is also no difference between the M4P and control IgG groups.

### 3.4. M4P Improves Vascular Integrity after 7-h Stroke Reperfusion

To understand whether blocking TRPM4 improves vascular integrity, we performed immunostaining on cerebral vasculature one day after occlusion in permanent MCAO animals and in 7-h MCAO reperfusion animals treated with M4P or control IgG (Figure 4A). The vascular diameter was quantified accordingly (Figure 4B). Both control IgG and M4P groups demonstrated a larger vascular diameter than the permanent MCAO. Between the control IgG and M4P groups, animals in the M4P group showed a larger diameter. This result indicates that M4P achieved better reperfusion than IgG.

Next, we compared the vascular integrity between the control IgG and M4P treatments post reperfusion by injecting Evans Blue dye. Extravasation of the dye stained the ipsilateral hemispheres in blue color (Figure 4C). Bleeding was also identified in the control IgG-treated rat brain, suggesting that 7-h stroke reperfusion-induced hemorrhage transformation in this rat. Quantification of Evans Blue dye showed that the leakage of dye was significantly reduced in the ipsilateral hemispheres of M4P-treated animals compared to the control IgG group (Figure 4D). In the contralateral hemispheres, there is no difference between the two treatments.

### 3.5. M4P on Cerebral Blood Flow and Functional Recovery

To examine whether reperfusion results in improved blood flow, we used a Laser-Doppler flowmetry to monitor the blood flow to the ipsilateral hemispheres at baseline, after MCAO, and after reperfusion (Figure 5A,B). The reperfusion was achieved by removing the filament from the middle cerebral artery at 7 h after occlusion. As illustrated by Figure 5A–C, the blood flow following filament withdrawal does not resume to the baseline level. The blood flow in the M4P group (43%) is significantly higher than the control IgG group (30.2%). We also quantified the blood flow in an early stroke reperfusion model in which the MCAO time was maintained for 3 h, and the antibodies were administered at 2 h post occlusion (Figure 5D). Compared to the 7-h transient MCAO, 3-h stroke reperfusion achieved a much higher blood flow resumption. In the M4P group, the blood flow to the ipsilateral hemisphere was elevated to 89.4% of baseline after recanalization, and from the control IgG group, the blood flow was resumed to 64.3%. Again, the blood flow of the M4P group was significantly higher than the control IgG group.

Next, we used the Rotarod test to assess the motor functions of animals after 7-h stroke reperfusion (Figure 5E). In sham-operated animals, the motor functions slightly dropped on day 1 post-operation. In the permanent MCAO group and 7-h stroke reperfusion groups treated with M4P or control IgG, motor functions dropped significantly on day 1 post-operation. The motor functions gradually recovered in the following days. However, we did not observe any difference among the three groups of animals receiving MCAO.

## 4. Discussion

Antibodies have been proposed for stroke therapy [44]. To our knowledge, TRPM4 blocking antibody M4P is the first antibody developed to target an ion channel for stroke therapy. M4P was shown to inhibit the TRPM4 channel function [10]. By binding to an extracellular domain close to the channel pore, M4P inhibits sodium influx and attenuates oncotic cell death in neurons and vascular endothelial cells under hypoxia [15]. In a 3-h transient MCAO animal model, M4P was revealed to ameliorate reperfusion injury and improve functional outcomes [10]. Here, we examined the role of M4P in a delayed 7-h stroke reperfusion animal model. In this model, M4P was given 1 h before filament removal at 7 h post occlusion. We have shown previously that antibodies could reach the occluded blood vessels, possibly via collateral circulation, when injected 1 h prior to recanalization [10]. This 1-h interval was selected based on the clinical observations from stroke patients receiving tPA, most of whom achieved recanalization within 60 min of tPA treatment [45]. This experimental design is to mimic the clinical scenario in which the antibody is proposed to be delivered together with the reperfusion drug tPA. Our study contrasts with other reports using putative neuroprotective agents administered either pre-ischemia, intra-ischemia, or shortly after reperfusion which are irrelevant to clinical situations [46]. Using this clinically relevant animal stroke model, we hypothesized that M4P has the potential to extend the time window of reperfusion therapy when applied together with tPA. Such a hypothesis was supported by the observation that, albeit with a structural loss, vasculature at 6 h post occlusion has a better morphology than the 9 h MCAO brain (Figure 1). Therefore, it is possible to achieve vascular protection at 6 h before the damage reaches a maximum at 9 h. The key question is whether the reperfusion injury induced by delayed reperfusion exceeds the beneficial effect obtained from reperfusion.

Compared to IgG, M4P treatment at 6 h post occlusion improved vascular integrity manifested by a larger vascular diameter after recanalization and a less severe leakage of Evans blue dye. Vascular endothelial swelling often occurs after ischemic onset, further narrowing the occluded blood vessels [4]. M4P has been found to attenuate endothelial swelling [15], which may facilitate perfusion when recanalization is successful. The overall result of vascular protection by M4P leads to a reduction of infarct volume and mortality rate during the delayed reperfusion. However, we did not observe a difference in functional recovery assessed by the Rotarod test. A possible reason is that the infarct volume was quantified 24 h after operation. In contrast, the functional test was performed throughout the whole process of stroke recovery. It should be noted that in the control IgG group, 23.5% of mortality occurred after 24 h. This figure reduces to 12.5% in the M4P group. Animals that died after 24 h may present with a larger infarct volume not captured by the TTC staining. Whereas in the functional study, only animals that completed the full course of tests were included for analysis. The delayed animal death may be caused by the expansion of lesions after recanalization, which has been reported in patients having received reperfusion therapy [47]. Most expansions of infarction were found largely within the reperfusion region. However, in a small portion of patients, the lesion can reach outside the reperfusion area. The possible causes of lesion expansion include newly formed microvascular dysfunction or cortical spreading depression, which warrants further study [47]. Such infarct growth has also been identified in animal models of permanent MCAO [48].

Reopening the occluded blood vessels sometimes does not achieve complete tissue reperfusion, referred to as reperfusion failure. Although the underlying mechanisms are not fully understood, some have been proposed, including capillary constricting and stalling with neutrophils and pericytes, large vessel constriction, microvascular occlusion with microclots, among others. [49]. In this study when the filament was removed at 7 h post-MCAO, we observed a moderately higher blood flow in both M4P and control IgG groups. However, blood reflow upon recanalization is much higher in the 3-h transient MC than in the 7-h transient MCAO. Since the surgical procedures are similar in both models, injury to the vascular wall during operation alone is unlikely to cause the difference. In contrast, the prolonged time of obstruction might be the determining factor. After 7 h occlusion, the vascular structure affected by hypoxia is more severely injured than the 3-h MCAO model. Thus, fewer blood vessels remained intact following filament withdrawal to be reperfused completely and successfully. A second possible reason is that a blood clot may form in the distal part of the blood vessel after prolonged occlusion. Once reperfusion is achieved, this blood clot may travel downstream and occlude distal branches.

## 5. Limitations and Future Directions

The major limitation or challenge of this study is the prolonged 7-h operational procedure and anesthesia. Although ketamine/xylazine was selected for its safety, ease of administration, and low mortality [50,51], we encountered a high mortality rate even in the sham-operated group. The 8.3% mortality rate (2/24) is close to the 10% permanent MCAO model (2/20), suggesting that prolonged anesthesia is harmful to the animals. Furthermore, repeated operational procedures to the carotid arteries and branches cause additional injuries. The potential for clot formation after 7 h of occlusion also confounded the efficiency of recanalization. We have tried to use tPA in the experiment to dissolve blood clots. However, our attempts failed due to the bleeding disorder during/after the operation. To overcome the limitations of MCAO, less invasive animal models of stroke, such as transcranial occlusion and photothrombosis models [52], can be considered to test the effect of M4P on functional outcomes.

As M4P is a polyclonal antibody against rodent TRPM4, it cannot be used in humans. We are now in the process of developing a humanized antibody against human TRPM4. In addition to reducing reperfusion injury, TRPM4-blocking antibodies can be examined in conjunction with other treatments such as antiplatelets [53], none invasive transcranial stimulation [54,55], and other neuroprotectants [56,57]. Better neuroprotection may be achieved with multiple therapies being used together.

## 6. Conclusions

We provide evidence showing that TRPM4 blocking antibody M4P can be an effective vascular protective agent in delayed stroke reperfusion. Although it has been reported that reperfusion at 4 h and 12 h does not change infarct volume formation [58], our results suggest that with proper vascular protection, reperfusion injury at a time point longer than 4 h can be alleviated. Although an improvement in vascular integrity and reduced mortality are evidenced, the functional outcomes following M4P treatment require further studies with a lesser invasive animal model. The translation of animal study to clinical practice needs to consider the anatomical and functional differences between human and animal brains [59]. Despite all the challenges, our study provides a novel approach to vascular protection that has the potential to extend the current reperfusion time window of tPA. Furthermore, TRPM4 blocking antibody could benefit patients receiving mechanical thrombectomy, which is performed at a later point than tPA [2]. As a co-therapy, TRPM4 blocking antibody certainly could improve the outcome of thrombectomy [60].

## Figures and Tables

**Figure 1 biomedicines-11-01480-f001:**
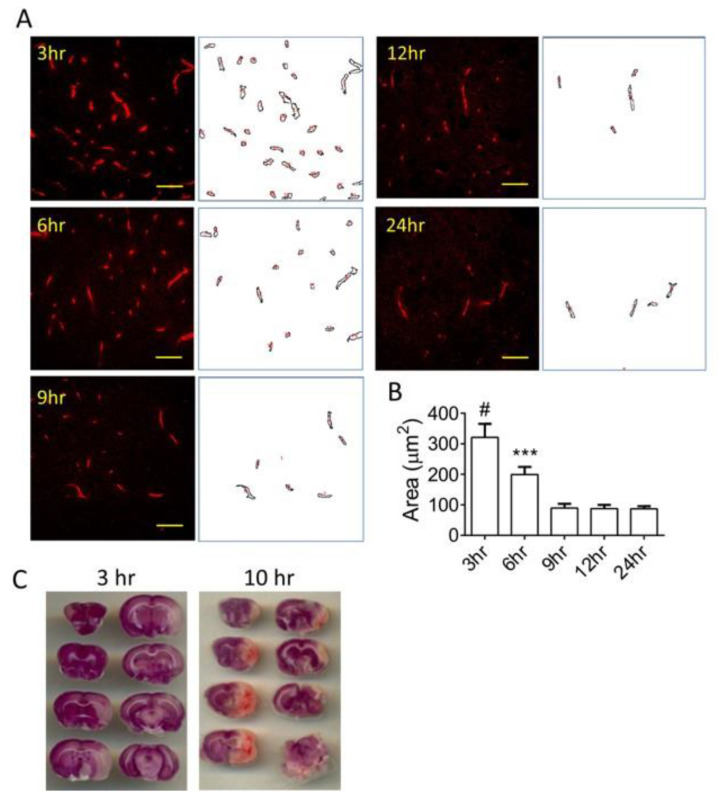
Time-dependent changes in the vasculature during stroke. (**A**) Representative images of cerebral vasculature within the ipsilateral hemispheres were captured at different time points after stroke induction. Corresponding ImageJ-processed vascular images were shown next to the immunofluorescent staining images. Scale bars: 50 μm. (**B**) Summary of vascular areas. *n* = 6–9 images from 3 rats. (**C**) Representative images of TTC staining obtained from a 3-h reperfusion rat brain and a 10-h reperfusion rat brain. Statistical analysis was performed by one-way ANOVA with Bonferroni’s post hoc analysis. *** *p* < 0.001, # *p* < 0.0001. The numerical data supporting the graphs can be found in Appendix A.

**Figure 2 biomedicines-11-01480-f002:**
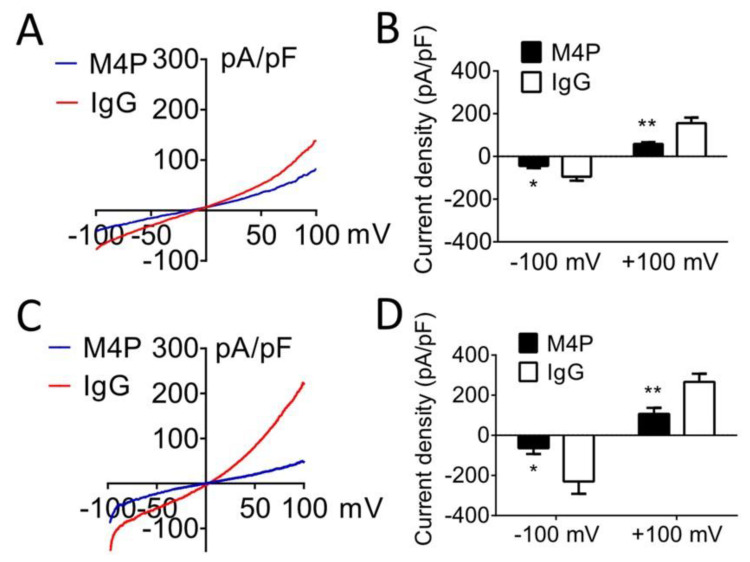
Inhibitory effect of TRPM4 blocking antibody M4P. (**A**) Exemplary current-voltage relationships of TRPM4 channel under the treatment of 20 µg/mL control IgG or 20 µg/mL M4P. Ramp protocols were applied from − 100 to + 100 mV with a holding potential at 0 mV. (**B**) Summary of current density at −100 and +100 mV. M4P: *n* = 12 cells; control IgG: *n* = 15 cells. (**C**) Exemplary current–voltage relationships of TRPM4 channel after 7-min ATP depletion. (**D**) Summary of current density under 7-min ATP depletion at −100 and +100 mV. M4P: *n* = 9 cells; control IgG: *n* = 12 cells. Statistical analysis was performed by student’s *t*-test. * *p* < 0.05, ** *p* < 0.01. The numerical data supporting the graphs can be found in Appendix A.

**Figure 3 biomedicines-11-01480-f003:**
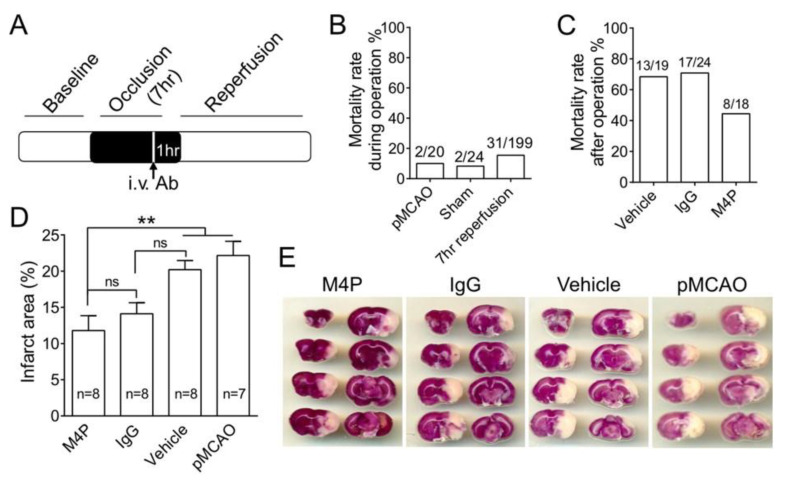
Evaluation of blocking TPRM4 in 7-h stroke reperfusion. (**A**) Diagram showing the experimental protocol for 7-h transient MCAO. M4P (100 µg) or control IgG (100 µg) was injected intravenously 1 h before recanalization. (**B**) Mortality rates during operation. Sham-operated animals received the same operation procedure as the 7-h transient MCAO except for the dissection of arteries and insertion of filaments. (**C**) Mortality rates after recovery from the operation. All 3 groups of animals underwent 7-h transient MCAO and were observed for 14 days post-operation. Vehicle group received i.v. injection of IgG elution buffer. (**D**) Summary of infarct area 24 h after operation. (**E**) Images of TTC-stained rat brains receiving permanent MCAO or 7-h transient MCAO with the treatments of M4P, control IgG, and vehicle. Statistical analysis was performed by one-way ANOVA with Bonferroni’s post hoc analysis. ** *p* < 0.01. The numerical data supporting the graphs can be found in Appendix A.

**Figure 4 biomedicines-11-01480-f004:**
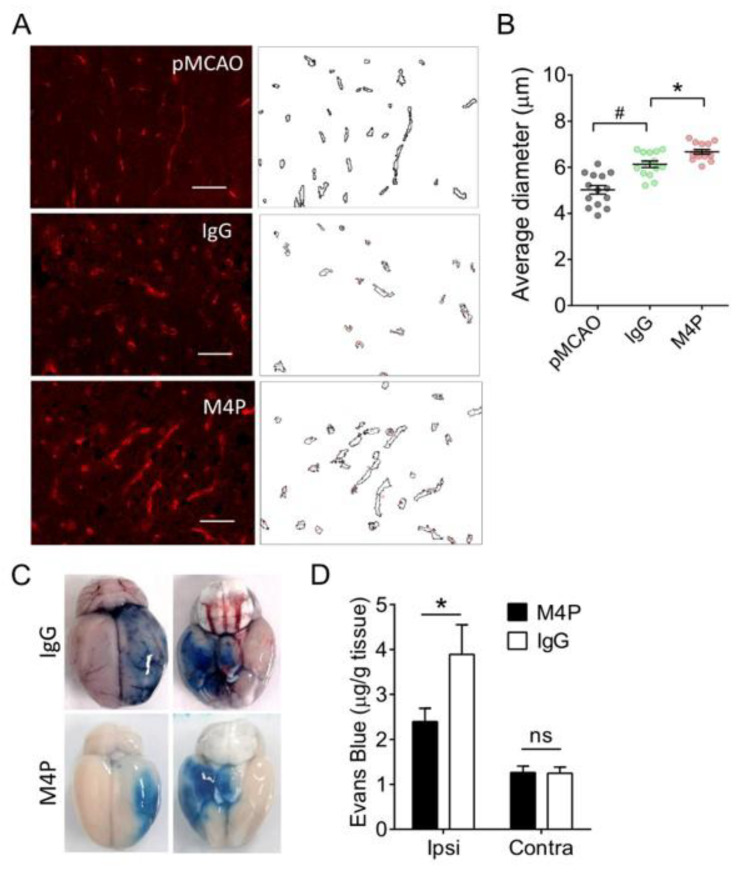
M4P improves vascular integrity during 7-h stroke reperfusion. (**A**) Representative images of immunofluorescent staining and corresponding ImageJ-processed images. 7-h transient MCAO rats receiving 100 µg control IgG or 100 µg M4P compared with permanent MCAO (pMCAO) without treatment. Scale bars: 50 μm. (**B**) Summary of the vascular diameter calculated by the shortest Feret diameter. In each group, *n* = 14 images were taken from 3 rats. (**C**) Dorsal and ventral views of sample rat brains with Evens blue staining. The rat brains were collected 24 h after the operation. (**D**) Summary of Evans blue quantification from ipsilateral and contralateral hemispheres. For M4P, *n* = 10 rats; for IgG, *n* = 8 rats. Statistical analysis was performed by one-way ANOVA with Bonferroni’s post hoc analysis for (**B**) and student’s *t*-test for (**D**). * *p* < 0.05, # *p* < 0.0001. The numerical data supporting the graphs can be found in Appendix A.

**Figure 5 biomedicines-11-01480-f005:**
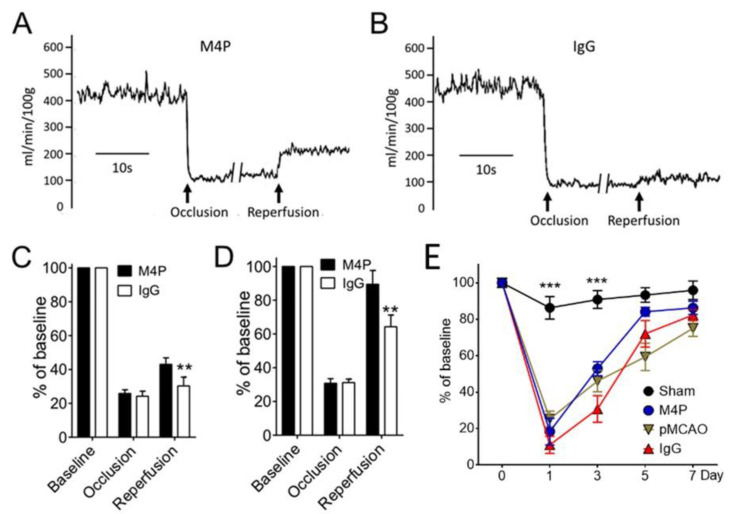
Functional analysis of M4P in 7-h stroke reperfusion. (**A**) Exemplary cerebral blood flow from a rat receiving 7-h MCAO. M4P of 100 µg was injected intravenously 1 h prior to recanalization. The blood flow was recorded using a Laser-Doppler flowmetry. (**B**) Exemplary cerebral blood flow from a rat treated with 100 µg control IgG. (**C**) Summary of cerebral blood flow from M4P and IgG-treated animals. The blood flow was normalized to baseline. For M4P, *n* = 9 rats; for IgG, *n* = 6 rats. (**D**) Cerebral blood flow from animals receiving 3-h stroke reperfusion. M4P (100 µg) and IgG (100 µg) were injected 1 h prior to recanalization. For M4P, *n* = 6 rats; for IgG, *n* = 10 rats. (**E**) Assessment of motor functions using the Rotarod test (*n* = 7 rats/group). Statistical analysis was performed by two-way ANOVA with Bonferroni’s post hoc analysis. ** *p* < 0.01, *** *p* < 0.001. The numerical data supporting the graphs can be found in Appendix A.

## Data Availability

All data generated and analyzed for this study are included in this published article.

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
