# Peer review of "TRPM4 Blocking Antibody Protects Cerebral Vasculature in Delayed Stroke Reperfusion"

_biomedicines, 2023, doi:10.3390/biomedicines11051480_

Round 1

Reviewer 1 Report

Chen and colleagues in the present article entitled ‘TRPM4 blocking antibody protects cerebral vasculature in a clinically relevant 7-hour stroke reperfusion animal model’, investigated the therapeutic effect of M4P during delayed stroke reperfusion.

In general, I think the idea of this article is really interesting and the authors’ fascinating observations on this timely topic may be of interest to the readers of Biomedicines. However, some comments, as well as some crucial evidence that should be included to support the author’s argumentation, needed to be addressed to improve the quality of the manuscript, its adequacy, and its readability prior to the publication in the present form. My overall judgment is to publish this paper after the authors have carefully considered my suggestions below, in particular reshaping parts of the ‘Introduction’ and ‘Methods’ sections by adding more evidence.

 Please consider the following comments:

I suggest changing the title. In my opinion, in the present form it is too wordy and it seems to be not enough clear and specific. 

Abstract: According to the Journal’s guidelines, the abstract should be a total of about 200 words maximum and should be presented as a single paragraph, without sub-headings. Please correct the actual one. Also, in my opinion, Authors should consider rephrasing this section. According to the Journal’s guidelines, the Abstract should contain most of the following kinds of information in brief form. Please, consider giving a more synthetic overview of the paper's key points: I would suggest rephrasing the results and conclusion to make them clear for readers to understand.

A graphical abstract that will visually summarize the main findings of the manuscript is highly recommended.

In general, I recommend authors to use more references to back their claims, especially in the Introduction of this research article, which I believe is lacking. Thus, I recommend the authors to attempt to expand the topic of their article, as the bibliography is too concise. Nevertheless, I believe that less than 60/70 articles are too low for a research article. Therefore, I suggest the authors to focus their efforts on researching relevant literature: in my opinion, adding more citations will help to provide better and more accurate background to this study. 

Results: I suggest rewriting this section more accurately. To properly present experimental findings, I think that authors should provide full statistical details (like degree of freedom or post-hoc utilized), to ensure in-depth understanding and replicability of the findings.

Discussion: In this final section, authors described the results of their study and their argumentation and captured the state of the art well; however, I would have liked to see some views on a way forward. I believe that the authors should make an effort, trying to explain the theoretical implication as well as the translational application of this paper, to adequately convey what they believe is the take-home message of their study. In this regard, I believe that it would be necessary to discuss theoretical and methodological avenues in need of refinement, as well as new suggestion of possible alternative and complementary treatments for post-stroke impairments, which may include the use of non-invasive brain stimulation techniques or neuropharmacological interventions (https://doi.org/10.1016/j.neubiorev.2023.105163; https://doi.org/10.3390/ijms24065926)

In my opinion, I think the ‘Conclusions’ paragraph would benefit from some thoughtful as well as in-depth considerations by the authors, because as it stands, it is very descriptive but not enough theoretical as a discussion should be. Authors should make an effort, trying to explain the theoretical implication as well as the translational application of their research.

I would ask the authors to include a proper and defined ‘Limitations and future directions’ section before the end of the manuscript, in which authors can describe in detail and report all the technical issues brought to the surface.

Figures: I suggest to modify all figures for clarity because, as it stands, the readers may have difficulty comprehending it. Also, please change the scale of the vertical axis and use the same minimum/maximum scale value in all the graphs.

References: Authors should consider revising the bibliography, as there are several incorrect citations. Indeed, according to the Journal’s guidelines, they should provide the abbreviated journal name in italics, the year of publication in bold, the volume number in italics for all the references. 

I hope that, after these careful revisions, this paper can meet the Journal’s high standards for publication. 

I am available for a new round of revision of this article. 

Best regards,

Reviewer

Minor editing of English language is required.

Author Response

Thanks for reviewer’s comment, to improve the manuscript, we make changes as suggested.

  • Title has been changed
  • The abstract has been modified to better describe our findings. The word count now is increased to 225.
  • A graphical abstract was made to summarize the clinical implications of our findings.
  • We also significantly increased the number of the references, including the relevant ones mentioned by the reviewers.
  • Statistical analyses are now described in more details in the figure legends and supplementary tables including post hoc methods used.
  • Introduction has been expanded to include more in-depth description on the background. Now the word count has increased to 949.
  • We have harmonized the vertical axis of Fig.3 for better reading.
  • Discussion part was improved with more clinical relevance. A new section of limitation and future directions are included.

Reviewer 2 Report

This study aimed to investigate the potential of the TRPM4 blocking antibody M4P in alleviating reperfusion injury and improving functional outcomes in a clinically relevant delayed 7-hour transient middle cerebral artery occlusion (MCAO) rat model. While the authors have provided some interesting findings, there are certain methodological limitations that hinder the strength of their conclusions.

The results indicate that M4P treatment improved vascular integrity and reduced infarct volume and mortality rate. These findings suggest that M4P could hold promise in stroke reperfusion therapy. However, the lack of significant improvement in functional outcome, as assessed by the Rotarod test, raises concerns about the overall effectiveness of M4P in a clinical setting. It is also important to note that the prolonged anesthesia and surgical procedure used in the study may have influenced the data analysis, further weakening the conclusions that can be drawn from the results.

While the study's focus on a delayed 7-hour transient MCAO model is commendable for its clinical relevance, the potential impact of prolonged anesthesia and surgical procedures on data analysis should have been better addressed. The authors acknowledge these limitations but fail to provide sufficient details on how they might have affected the results or how they could be overcome in future research.

Moreover, the lack of significant improvement in functional outcomes calls into question the overall therapeutic potential of M4P. In future studies, it would be beneficial to explore alternative functional assessment methods and to optimize the experimental design to better understand the true impact of M4P on functional recovery.

The manuscript is well-written, but some revisions are recommended. The authors emphasize the importance of early reperfusion, focusing primarily on tPA-based pharmacological reperfusion. However, the impact of reperfusion injury may differ between pharmacological and mechanical reperfusion, considering the lower rate of unsuccessful reperfusion and differences in timing. Future medical therapies should target the enhancement of adjuvant pharmacotherapy in the context of modern mechanical reperfusion, addressing its unmet needs. Moreover, it is crucial to discuss potential interactions with the existing adjuvant treatments, including anticoagulant and antiplatelet medications (as recently reviewed in PMID: 35976963).

In conclusion, this study offers intriguing findings regarding the potential of the TRPM4 blocking antibody M4P in ameliorating vascular injury during delayed stroke reperfusion. However, the methodological limitations and the lack of significant improvement in functional outcomes warrant caution in interpreting the results. Further research is needed to refine the experimental design and address the limitations of this study to better assess the potential of M4P as a treatment for stroke reperfusion injury.

Author Response

 Thanks for reviewer’s comment, to improve the manuscript, we make changes as suggested.

  • We added more details in introduction and discussion.
  • A new section of limitation and future directions are added, including the discussion of potential anticoagulant and antiplatelet medications.